# Geospatial epidemiology of hospitalized patients with a positive influenza assay: A nationwide study in Iran, 2016–2018

Shahab MohammadEbrahimi[1,2], Behzad Kiani ©[3]*, Zahra Rahmatinejad[1], Stefan Baral[4], Soheil Hashtarkhani[5], Mohammad Dehghan-Tezerjani[6], Elahe Zare[7], Mahnaz Arian[8]*, Fatemeh Kiani[9], Mohammad Mehdi Gouya[10], Mohammad Nasr Dadras[10], Mohammad Karamouzian[11,12,13]

1 Department of Medical Informatics, School of Medicine, Mashhad University of Medical Sciences, Mashhad, Iran, 2 Student Research Committee, Mashhad University of Medical Sciences, Mashhad, Iran, 3 École de Santé Publique de L'Université de Montréal (ESPUM), Québec, Montréal, Canada, 4 Department of Epidemiology, Johns Hopkins Bloomberg School of Public Health, Baltimore, Maryland, United States of America, 5 Department of Health Information Technology, Neyshabur University of Medical Sciences, Neyshabur, Iran, 6 Department of Anesthesiology and Critical Care, Shahid Sadoughi University of Medical Sciences, Yazd, Iran, 7 Cardiovascular Research Center, Shahid Sadoughi University of Medical Sciences, Yazd, Iran, 8 Department of Infectious Diseases, School of Medicine, Mashhad University of Medical Sciences, Mashhad, Iran, 9 Department of Health Information Technology and Management, School of Allied Medical Sciences, Shahid Beheshti University of Medical Sciences, Tehran, Iran, 10 Center for Communicable Disease Control, Ministry of Health and Medical Education, Tehran, Iran, 11 Brown School of Public Health, Brown University, Providence, RI, United States of America, 12 Centre on Drug Policy Evaluation, St. Michael's Hospital, Toronto, ON, Canada, 13 HIV/STI Surveillance Research Center, and WHO Collaborating Center for HIV Surveillance, Institute for Futures Studies in Health, Kerman University of Medical Sciences, Kerman, Iran

* kiani.behzad@gmail.com (BK); arianm@mums.ac.ir (MA)

**Editor:** Hamid Sharifi, HIV/STI Surveillance Research Center and WHO Collaborating Center for HIV Surveillance, Institute for Future Studies in Health, Kerman University of Medical Sciences, ISLAMIC REPUBLIC OF IRAN

## Abstract

### Introduction

Seasonal influenza is a significant public health challenge worldwide. This study aimed to investigate the epidemiological characteristics and spatial patterns of severe hospitalized influenza cases confirmed by polymerase chain reaction (PCR) in Iran.

### Methods

Data were obtained from Iran's Ministry of Health and Medical Education and included all hospitalized lab-confirmed influenza cases from January 1, 2016, to December 30, 2018 (n = 9146). The Getis-Ord Gi* and Local Moran's *I* statistics were used to explore the hotspot areas and spatial cluster/outlier patterns of influenza. We also built a multivariable logistic regression model to identify covariates associated with patients' mortality.

### Results

Cumulative incidence and mortality rate were estimated at 11.44 and 0.49 (per 100,000), respectively, and case fatality rate was estimated at 4.35%. The patients' median age was 40 (interquartile range: 22–63), and 55.5% (n = 5073) were female. The hotspot and cluster analyses revealed high-risk areas in northern parts of Iran, especially in cold, humid, and

**Data Availability Statement:** All relevant data are within the manuscript and its Supporting Information files (S2 Appendix).

**Funding:** This study was financially funded by Mashhad University of Medical Sciences (fund number: 4000963). B.K awarded the fund. The funder did not play any role in the study design and preparation of the manuscript.

**Competing interests:** The authors have declared that no competing interests exist.

densely populated areas. Moreover, influenza hotspots were more common during the colder months of the year, especially in high-elevated regions. Mortality was significantly associated with older age (adjusted odds ratio [aOR]: 1.01, 95% confidence interval [CI]: 1.01–1.02), infection with virus type-A (aOR: 1.64, 95% CI: 1.27–2.15), male sex (aOR: 1.77, 95% CI: 1.44–2.18), cardiovascular disease (aOR: 1.71, 95% CI: 1.33–2.20), chronic obstructive pulmonary disease (aOR: 1.82, 95% CI: 1.40–2.34), malignancy (aOR: 4.77, 95% CI: 2.87–7.62), and grade-II obesity (aOR: 2.11, 95% CI: 1.09–3.74).

## Conclusions

We characterized the spatial and epidemiological heterogeneities of severe hospitalized influenza cases confirmed by PCR in Iran. Detecting influenza hotspot clusters could inform prioritization and geographic specificity of influenza prevention, testing, and mitigation resource management, including vaccination planning in Iran.

## Introduction

Out of the estimated one billion cases of influenza annually worldwide, up to five million are severe (about half of one percent of the total estimate), with more than 500,000 infections expected to result in death [1]. Seasonal influenza epidemics impose a significant burden on the healthcare system of low- and middle-income countries (LMICs) around the world [2, 3]. In the eastern Mediterranean region (EMR), influenza outbreaks represent a potential threat for a global pandemic due to an array of regional, cultural, and structural influences, such as fragile health systems, inadequate disease surveillance, rapid urbanization, climate fluctuations, and increased human-animal interaction due to its proximity to the migratory pathways of birds [4, 5]. However, the socio-economic burden of influenza and its substantial morbidity and mortality remain underappreciated in the EMR, highlighting the need for further efforts to improve influenza surveillance, monitoring, and response measures [6].

Iran is one of the countries in EMR with a significant burden of influenza. According to the global burden of disease estimations in 2017, the influenza burden in Iran had an incidence and mortality rate of 587/100,000 people and 0.8/100,000 people, respectively [6]. Based on a systematic review in Iran, influenza prevalence varied greatly between 1.3% to 52% in different populations (all people, adults, or children) and areas. Furthermore, the most prevalent influenza subtype is the H1N1 which is also associated with the highest mortality rate compared to other subtypes [7]. The high burden of influenza in Iran could also be attributed to low immunization rates, particularly among higher-risk older adults and those affected by severe comorbidities [8].

From a geographical point of view, spatiotemporal factors represent among the most important and influential factors in the spread of influenza [9, 10]. Since adjacent areas have similar spatial and temporal characteristics, high-risk areas could be detected based on space-time correlations [11]. The burden of diseases is not distributed randomly in a specific area and fluctuates by location [12]. Indeed, the burden of influenza fluctuates by location, and its distribution could be driven by diffusion patterns through larger population hubs to surrounding communities [13, 14]. Geographical Information System (GIS) is a useful tool to visualize space-time information and can be considered as a decision support system [15]. In Iran, several epidemiological studies on influenza have been conducted [7]; however, little attention has been paid to the spatial correlation of adjacent areas and exploration of hotspot clusters [16, 17].

While an international body of evidence consistently demonstrates how spatial and epidemiological factors impact the seasonality of influenza incidence, the understanding of spatial distribution of influenza remains limited in Iran. Even less is known about hospitalized cases of influenza across the country. Identifying high-risk clusters of influenza could help Iran's monitoring and therapeutic measures. Therefore, we applied geospatial and statistical approaches to explore spatial patterns and epidemiological characteristics of hospitalized influenza cases confirmed by polymerase chain reaction.

## Methods

### Study design and setting

We conducted a retrospective study to explore geospatial patterns of seasonal influenza in Iran. Iran is located in northeast of Persian Gulf, with latitude and longitude of 32°00' N and 53°00' E, and has 31 provinces, 388 counties, and 1245 cities.

### Data sources

Data were obtained from the Ministry of Health and Medical Education's Center for Communicable Disease Control (CCDC) database from January 1, 2016, to December 31, 2018. Data included all inpatient cases hospitalized due to influenza in this period, confirmed by a positive laboratory test using the real-time reverse transcription-polymerase chain reaction (RT-PCR) assay (n = 9146). This surveillance data included demographic information (sex, age, primary/ permanent residence address), type of pathogen agent (A-type [H1N1, H3N2], or B-type), manifested systemic symptoms, self-reported severe comorbidities, and the final health outcome (death or recovery).

### Data analysis

**Disease mapping.** The data were geocoded at the county level. The cumulative incidence of influenza infections was calculated using the average population of each province and county, according to Iran's population and housing censuses data (2016) [18]. A cartographic map of influenza cumulative incidence was generated using natural break classification with five classes. This method seeks to reduce the within-class variance and maximize the between-class variance [19]. Depending on the location of the study area, the projection system of WGS_1984_World_Mercator was used for projecting the GIS layers.

### Statistical analysis

**Descriptive statistics and multivariable regression.** Categorical variables were expressed as numbers (%), and continuous variables were presented as median and interquartile range (IQR). The continuous measures were compared using the Mann-Whitney U test, while dichotomous data were compared using $\chi^2$ or Fisher's exact test, as appropriate. Demographic characteristics, comorbidities (pre-existing conditions), manifested symptoms, and infectious virus type/subtype were analyzed annually and overall. To determine the factors independently associated with mortality among influenza patients, univariable and multivariable logistic regression models were built. Variables with a $p < 0.05$ in the univariable model were entered into the multivariable model, which is one of the approaches that can be used to choose what variables should be included in regression analysis [20]. This filtering system of variables helps to avoid adding extra variables in the logistic regression which can cause an unrealistic model. The final model was selected using a backward stepwise regression approach. Since the way variables are encoded is very influential in interpreting the results of regression models; virus type-B, female sex and the absence of comorbidities were considered as references for our

model. Adjusted odds ratio (aOR) along with a 95% confidence interval (CI) were reported. Age-specific rates, including cumulative incidence, mortality rate (MR), and case fatality rate (CFR), were calculated.

**Hotspot analysis (Getis-Ord Gi\*).** A hotspot is an area within a prescribed limit with concentration or dispersion of occurrences of the same value [21]. The Getis-Ord Gi* statistic detects the presence of local spatial autocorrelations. To be a statistically significant hotspot, the tract (region) needs to be surrounded by tracts (regions) with high, positive values and have a significantly higher, positive value than its neighbors. In a simple word: "The local sum for a feature and its neighbors is compared proportionally to the sum of all features; when the local sum is very different from the expected local sum, and that difference is too large to be the result of random choice, a statistically significant z-score results". The inverse is also true for the cold spots [22, 23]. We applied this statistic to identify hotspots and cold spots based on cumulative influenza incidences at the county as spatial features. This is similar to the high-high (HH) or low-low (LL) relationships that the Local Moran's *I* test detects [24]. However, Local Moran's *I* can also detect high-low (HL) or low-high (LH) relationships, whereas hotspot analysis just looks for clusters of similar high or low values.

**Cluster and outlier analysis (Local Moran's I).** The Local Moran's *I* was used to quantify spatial autocorrelation of cumulative incidences. Its value varies between -1 and +1 and determines whether the apparent similarity (a spatial clustering, either high or low) or dissimilarity (a spatial outlier) is pronounced or not [24]. In other words, the null hypothesis states that the influenza cases are randomly distributed. The analysis can detect four types of clusters: HH and LL areas indicate clusters of influenza occurrence, but the HL and LH areas indicate the outliers.

## Software and significance level

All tests were two-sided with $p<0.05$ considered statistically significant. All the descriptive maps and spatial analyses in this study were created by the authors using ArcGIS software, version 10.8. Non-spatial statistical analyses were performed using R software, version 4.2.1 (R Foundation for Statistical Computing), and Microsoft Excel 2016.

## Ethics statement

The Mashhad University of Medical Sciences ethical committee approved this study with the reference number of IR.MUMS.MEDICAL.REC.1400.629. All data were fully anonymized; therefore, the ethical committee waived the need for informed consent.

## Results

### Demographic characteristics

We analyzed 9146 confirmed influenza infections during three years in Iran. The total CFR for influenza patients was 4.35% (95% CI: 3.95–4.79), and the highest incidence (41.7, 95% CI: 40.7–42.7) and mortality (41.7, 95% CI: 37.1–46.8) rates belonged to the year 2016 (**Fig 1A**). **Table 1** provides a comprehensive comparison of the influenza cases concerning their responses to the influenza infection. The baseline characteristics of the patients were stratified by mortality. Median age of the patients was 40 years (IQR: 22–63), and over 55% (n = 5073) of them were female. Compared to females, males were significantly more likely to have died from influenza (57.3% vs. 42.7%). Moreover, the deceased patients were significantly older by median (i.e., 56 years, IQR: 34–74 vs. 39 years, IQR: 22–63). In addition, detailed information regarding cumulative incidence rate, mortality rate (MR), and case fatality rate (CFR), alongside comorbidities frequencies stratified by each province, are presented in **S1 Appendix**.

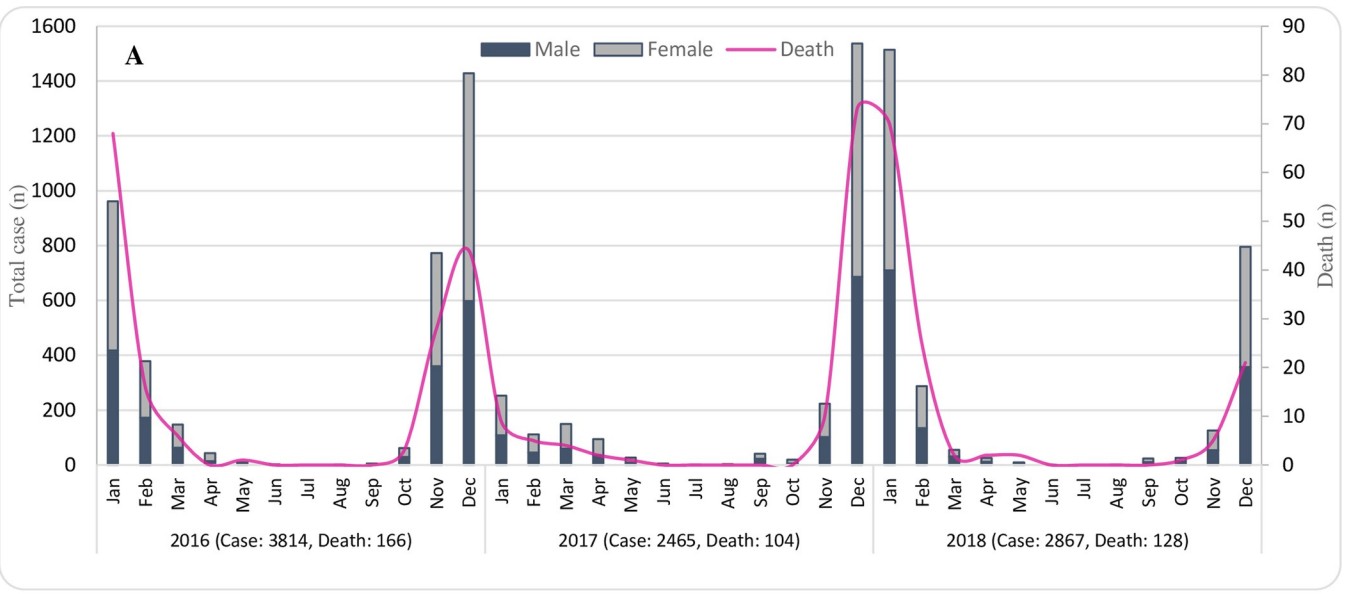

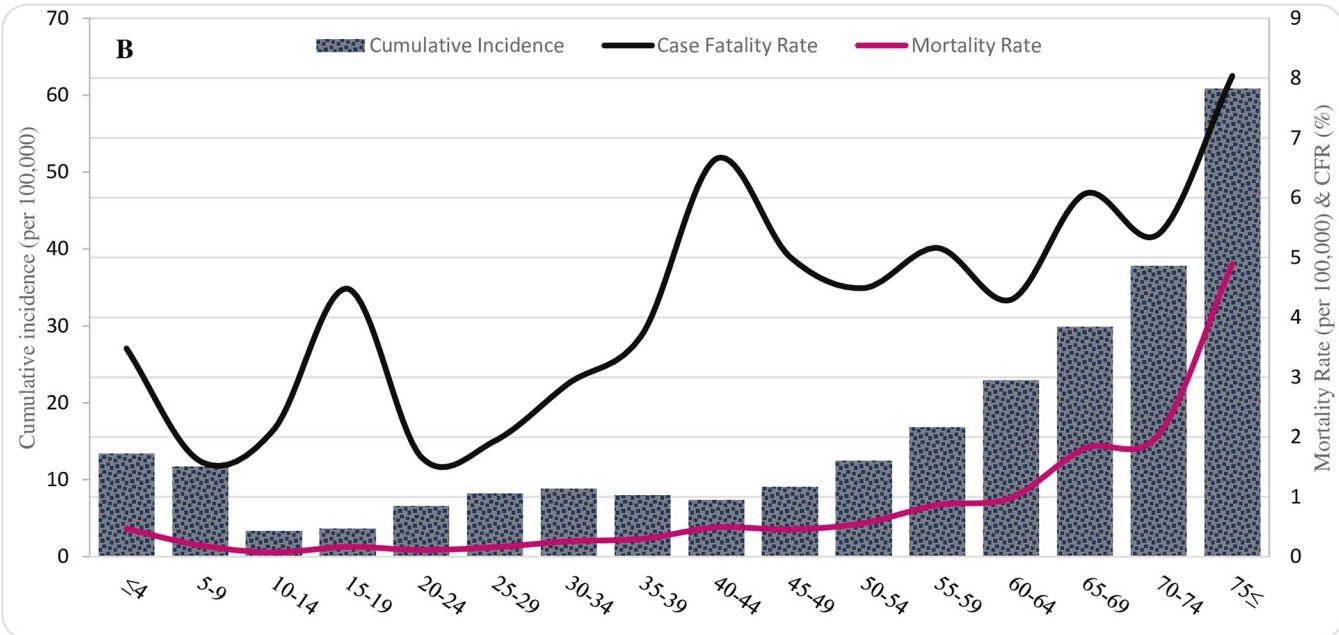

**Fig 1.** (A) Epidemiologic trend of confirmed influenza cases and mortality in Iran separated by sex, (B) Distribution of age-specific incidence, mortality, and case-fatality rates of influenza in Iran, 2016–2018.

## Age-specific rates

We categorized patients into 16 categories of five-year age intervals and calculated the age-specific cumulative incidence, age-specific MR, and age-specific CFR (Fig 1B). The population-based data categorized by age groups were obtained from Iran's population and housing census data [18]. The overall cumulative incidence and MR of influenza in Iran were 11.44 (95% CI: 11.21–11.68) and 0.49 (95% CI: 0.45–0.55) per 100,000 over the study period. The highest cumulative incidence (60.88 per 100,000, 95% CI: 57.55–64.42), MR (4.89 per 100,000, 95% CI:

**Table 1. Baseline characteristics stratified by mortality and survival of influenza patients in Iran, 2016–2018.**

| Demographic and Clinical Characteristics | Survivor | | | | Non-survivor | | | | Total | | | | p-value |
|---|---|---|---|---|---|---|---|---|---|---|---|---|---|
| | 2016 (3648) | 2017 (2361) | 2018 (2739) | total (8748) | 2016 (166) | 2017 (104) | 2018 (128) | total (398) | 2016 (3814) | 2017 (2465) | 2018 (2867) | total (9146) | |
| Age | 41.00 | 37.00 | 40.00 | 39.00 | 56.50 | 53.00 | 57.50 | 56.00 | 42.00 | 38.00 | 41.00 | 40.00 | <0.001[a] |
| (Age range) | 25.0–65.0 | 14.0–60.0 | 18.0–63.0 | 22.0–63.0 | 34.0–74.0 | 34.0–73.0 | 36.0–74.0 | 34.0–74.0 | 25.0–65.0 | 16.0–60.0 | 19.0–64.0 | 22.0–63.0 | |
| **Sex** | | | | | | | | | | | | | |
| Male | 1577 (43.22) | 1021 (43.25) | 1247 (45.53) | 3845 (43.95) | 89 (53.61) | 58 (55.76) | 81 (63.28) | 228 (57.29) | 1666 (43.68) | 1079 (43.77) | 1328 (46.32) | 4073 (44.53) | <0.001[b] |
| Female | 2071 (56.78) | 1340 (56.75) | 1492 (54.47) | 4903 (56.05) | 77 (46.39) | 46 (44.24) | 47 (36.72) | 170 (42.71) | 2148 (56.32) | 1386 (56.23) | 1539 (53.68) | 5073 (55.47) | |
| **Comorbidities** | | | | | | | | | | | | | |
| CVD | 526 (14.41) | 298 (12.62) | 386 (14.09) | 1,210 (13.83) | 50 (30.12) | 26 (25.0) | 34 (26.56) | 110 (27.63) | 576 (15.10) | 324 (13.14) | 420 (14.65) | 1,320 (14.43) | <0.001[b] |
| Diabetes | 301 (8.25) | 197 (8.34) | 236 (8.61) | 734 (8.39) | 22 (13.25) | 16 (15.38) | 21 (16.40) | 59 (14.82) | 323 (8.47) | 213 (8.64) | 257 (8.96) | 793 (8.67) | <0.001[b] |
| CRD | 125 (3.43) | 63 (3.12) | 97 (3.54) | 285 (3.26) | 11 (6.63) | 6 (5.77) | 8 (6.25) | 25 (6.28) | 136 (3.57) | 69 (2.80) | 105 (3.66) | 310 (3.39) | 0.001[b] |
| CLD | 35 (0.96) | 24 (1.02) | 23 (0.84) | 82 (0.94) | 5 (3.01) | 2 (1.92) | 2 (1.56) | 9 (2.26) | 40 (1.05) | 26 (1.05) | 25 (0.87) | 91 (0.99) | <0.001[b] |
| COPD | 509 (13.95) | 242 (10.25) | 356 (13.0) | 1107 (12.65) | 43 (25.90) | 21 (20.19) | 39 (30.47) | 103 (25.88) | 552 (14.47) | 263 (10.67) | 395 (13.78) | 1210 (13.23) | <0.001[b] |
| Malignancy | 46 (1.26) | 31 (1.31) | 31 (1.13) | 108 (1.24) | 10 (6.02) | 6 (5.77) | 6 (4.69) | 22 (5.53) | 56 (1.47) | 37 (1.50) | 37 (1.29) | 130 (1.42) | <0.001[b] |
| Obesity (grade-II)[c] | 41 (1.13) | 28 (1.19) | 39 (1.42) | 108 (1.24) | 4 (2.41) | 5 (4.81) | 4 (3.12) | 13 (3.26) | 45 (1.18) | 33 (1.34) | 43 (1.50) | 121 (1.32) | <0.001[b] |
| **Virus type** | | | | | | | | | | | | | |
| A | 3099 (84.95) | 1405 (59.50) | 1834 (66.96) | 6338 (72.45) | 151 (90.96) | 82 (78.85) | 92 (71.87) | 325 (81.66) | 3250 (85.21) | 1487 (60.32) | 1926 (67.17) | 6663 (72.85) | <0.001[b] |
| H1N1 | 704 (22.72) | 651 (46.33) | 405 (22.08) | 1760 (27.77) | 62 (41.06) | 33 (40.24) | 35 (38.04) | 130 (40.00) | 766 (23.57) | 684 (46.00) | 440 (22.84) | 1890 (28.37) | |
| H3N2 | 2216 (71.50) | 448 (31.89) | 991 (54.03) | 3655 (57.64) | 72 (47.68) | 19 (23.17) | 34 (36.96) | 125 (38.46) | 2288 (70.40) | 467 (31.40) | 1025 (53.22) | 3780 (56.73) | |
| Not subtyped | 179 (5.78) | 306 (21.78) | 438 (23.88) | 923 (14.56) | 17 (11.26) | 30 (36.59) | 23 (25.00) | 70 (21.54) | 196 (6.03) | 336 (22.60) | 461 (23.94) | 993 (14.90) | |
| B | 549 (15.05) | 956 (40.50) | 905 (33.04) | 2410 (27.55) | 15 (9.04) | 22 (21.15) | 36 (28.13) | 73 (18.34) | 564 (14.79) | 978 (39.67) | 941 (32.83) | 2483 (27.15) | |
| **Symptoms** | | | | | | | | | | | | | |
| Fever | 3068 (84.10) | 1973 (83.57) | 2150 (78.50) | 7191 (82.20) | 122 (73.49) | 74 (71.15) | 84 (65.63) | 280 (70.35) | 3190 (83.64) | 2047 (83.04) | 2234 (77.92) | 7471 (81.68) | <0.001[b] |
| Sore throat | 1258 (34.48) | 892 (37.78) | 930 (33.95) | 3080 (35.20) | 36 (21.69) | 27 (25.96) | 23 (17.97) | 86 (21.60) | 1294 (33.93) | 919 (37.28) | 953 (33.24) | 3166 (34.62) | <0.001[b] |
| Dyspnea | 1134 (31.08) | 750 (31.77) | 779 (28.44) | 2663 (30.44) | 25 (15.06) | 17 (16.34) | 15 (11.72) | 57 (14.32) | 1159 (30.39) | 767 (31.11) | 794 (27.69) | 2720 (29.74) | <0.001[b] |
| Hemoptysis | 1618 (44.35) | 1103 (46.72) | 1065 (38.88) | 3786 (43.28) | 54 (32.53) | 31 (29.81) | 32 (25.0) | 117 (29.40) | 1672 (43.84) | 1134 (46.0) | 1097 (38.27) | 3903 (42.67) | 0.001[b] |
| Chest pain | 751 (20.59) | 500 (21.18) | 549 (20.04) | 1800 (20.58) | 23 (13.85) | 11 (10.58) | 10 (7.81) | 44 (11.05) | 774 (20.29) | 511 (20.73) | 559 (19.50) | 1844 (20.16) | <0.001[b] |
| Pregnancy (Total females: 5,073) | 413/2071 (19.94) | 201/1340 (15.0) | 259/1492 (17.36) | 873/4903 (17.80) | 3/77 (3.90) | 2/46 (4.35) | 3/47 (8.51) | 8/170 (4.70) | 416/2148 (19.36) | 203/1386 (14.64) | 262/1539 (17.02) | 881/5073 (17.37) | <0.001[b] |
| Overseas travel [d] | 134 (3.67) | 115 (4.87) | 45 (1.64) | 294 (3.36) | 11 (6.63) | 7 (6.73) | 1 (0.78) | 19 (4.77) | 145 (3.80) | 122 (4.95) | 46 (1.60) | 313 (3.42) | 0.1294[b] |

(*Continued*)

**Table 1.** (Continued)

| Demographic and Clinical Characteristics | Survivor | | | | Non-survivor | | | | Total | | | | p-value |
|---|---|---|---|---|---|---|---|---|---|---|---|---|---|
| | 2016 (3648) | 2017 (2361) | 2018 (2739) | total (8748) | 2016 (166) | 2017 (104) | 2018 (128) | total (398) | 2016 (3814) | 2017 (2465) | 2018 (2867) | total (9146) | |
| Domestic travel [d] | 111 (3.04) | 55 (2.33) | 56 (2.04) | 222 (2.01) | 4 (2.41) | 2 (1.92) | 2 (1.56) | 8 (2.53) | 115 (3.01) | 57 (2.31) | 58 (2.02) | 230 (2.51) | 0.5108[b] |

[*] The reported p-values in the last column are related to the significant comparison of patients based on their health outcomes in total, i.e., survivors (n = 8748) vs. non-survivors (n = 398).

**a:** Mann-Whitney U test

**b:** Fisher's exact test ($\chi 2$)

**c:** Body mass index >35 kg/m2

**d:** Up to 7 days before admission.

CVD: cardiovascular disease; CRD: chronic renal disease; CLD: chronic liver disease; COPD: chronic obstructive pulmonary disease.

4.01–5.97), and CFR (8.04%, 95% CI: 6.64–9.72) were related to the age groups over 74-year-old age group. The lowest influenza cumulative incidence was observed among adolescents aged 10 to 19 (3.49 per 100,000, 95% CI: 3.16–3.85).

## Comorbidities and symptoms

As shown in **Table 1**, among the seven reported comorbidities, cardiovascular diseases (CVD) (14.4%, n = 1320), chronic obstructive pulmonary disease (COPD) (13.2%, n = 1210), and diabetes (8.7%, n = 793) were the most common comorbidities. The deceased patients were significantly more likely to have more severe comorbidities. Among the reported symptoms of influenza, fever was the most common (81.7%, n = 7471), followed by hemoptysis (42.7%, n = 3903) and sore throat (34.6%, n = 3166).

## Virus type

Regarding the type of influenza virus, type-A influenza was more prevalent than type-B virus (72.9% vs. 27.1%), and 81.9% (n = 325) of those who died were infected by the type-A virus. Among type-A subtypes, H3N2 was more contagious (56.73%, n = 3780) and H1N1 was more deadly (i.e., 6.9% (130/1890) of patients infected by the H1N1 subtype and 3.3% (125/3780) those infected by the H3N2 subtype passed away).

## Mortality risk factor analysis

The multivariable logistic regression model suggested mortality was significantly and positively associated with age (aOR: 1.01; 95% CI: 1.01–1.02) _ in a way that with each passing year of age, 0.01 will be added to the mortality chance_, infection with type-A virus (aOR: 1.64, 95% CI: 1.27–2.15), and male sex (aOR: 1.77; 95% CI: 1.44–2.18). Additionally, higher odds of mortality were observed among those with CVD (aOR: 1.71; 95% CI: 1.33–2.20), COPD (aOR: 1.82; 95% CI: 1.40–2.34), malignancy (aOR: 4.77, 95% CI: 2.87–7.62), and grade-II obesity (aOR: 2.11, 95% CI: 1.09–3.74) (**Table 2**).

## Spatial analysis

**Fig 2** shows the descriptive and hotspot analysis of influenza incidence at the county level, 2016–2018. **Fig 2B** is developed based on the Getis-Ord Gi* analysis, which is a hot spot analysis method and works by looking at each feature in the dataset within the context of

**Table 2. The univariable and multivariable logistic regression analysis of death in influenza patients.**

| Variable (Ref.) | Univariable OR | 95% CI | *p*-value | Multivariable OR ** | 95% CI | *p*-value |
|---|---|---|---|---|---|---|
| Age* | 1.02 | (1.01–1.02) | <0.0001 | 1.01 | (1.01–1.02) | <0.0001 |
| Virus type (B) A | 1.69 | (1.32–2.21) | <0.0001 | 1.64 | (1.27–2.15) | <0.0001 |
| Sex (Female) Male | 1.71 | (1.40–2.10) | <0.0001 | 1.77 | (1.44–2.18) | <0.0001 |
| CVD (No) Yes | 2.38 | (1.89–2.98) | <0.0001 | 1.71 | (1.33–2.20) | <0.0001 |
| Diabetes (No) Yes | 1.90 | (1.41–2.51) | <0.001 | - | - | - |
| CRD (No) Yes | 1.99 | (1.27–2.97) | 0.001 | - | - | - |
| COPD (No) Yes | 2.41 | (1.90–3.03) | <0.0001 | 1.82 | (1.40–2.34) | <0.0001 |
| CLD (No) Yes | 2.45 | (1.14–4.64) | 0.012 | - | - | - |
| Malignancy (No) Yes | 4.68 | (2.86–7.34) | <0.0001 | 4.77 | (2.87–7.62) | <0.0001 |
| Obesity-grade II (No) Yes | 2.70 | (1.44–4.67) | <0.001 | 2.11 | (1.09–3.74) | 0.017 |

*Treated as a continuous variable.

** Backward elimination approach was applied.

OR: odds ratio; CI: confidence interval; CVD: cardiovascular disease; CRD: chronic renal disease; COPD: chronic obstructive pulmonary disease; CLD: chronic liver disease. All comparisons are (yes vs. no) unless otherwise specified.

neighboring features in the same dataset. In order to be a significant hotspot, a feature (county) with a high value must be surrounded by other features (counties) with high values. The northern part of Iran including Tehran, the capital of Iran, experienced more influenza incident cases. Based on the hotspot analysis (**Fig 2B**), the hotspot areas were only detected in the northern parts of Iran, including Tehran and Karaj, two of the most populous cities of Iran as well as the cities neighboring the Caspian Sea.

**Fig 3** shows the descriptive monthly progression of influenza occurrence, using natural break classification. This map depicts the monthly cumulative incidence of influenza in different counties from 2016 to 2018. Each map contains three repetitions of each month's influenza cumulative incidences in 2016, 2017, and 2018. The cumulative incidence of influenza occurrence was higher from the end of autumn (November) to the beginning of spring (March), across the study period.

According to the Local Moran's I analysis, there was a spatial autocorrelation of the seasonal influenza at the county level. **Fig 4** shows the hotspot clustering based on monthly cumulative incidences. In the cold months (December to April) HH clusters were detected in the northern part of the country. Moreover, LL clusters were detected in the hot months and the highest frequency of these clusters was observed in July, the hottest month of the year.

## Discussion

We reviewed the spatial and epidemiological patterns of seasonal influenza confirmed by RT-PCR among hospitalized cases in Iran from January 1, 2016, to December 30, 2018. Based on our surveillance data, hotspot areas and HH clusters were identified in the north and

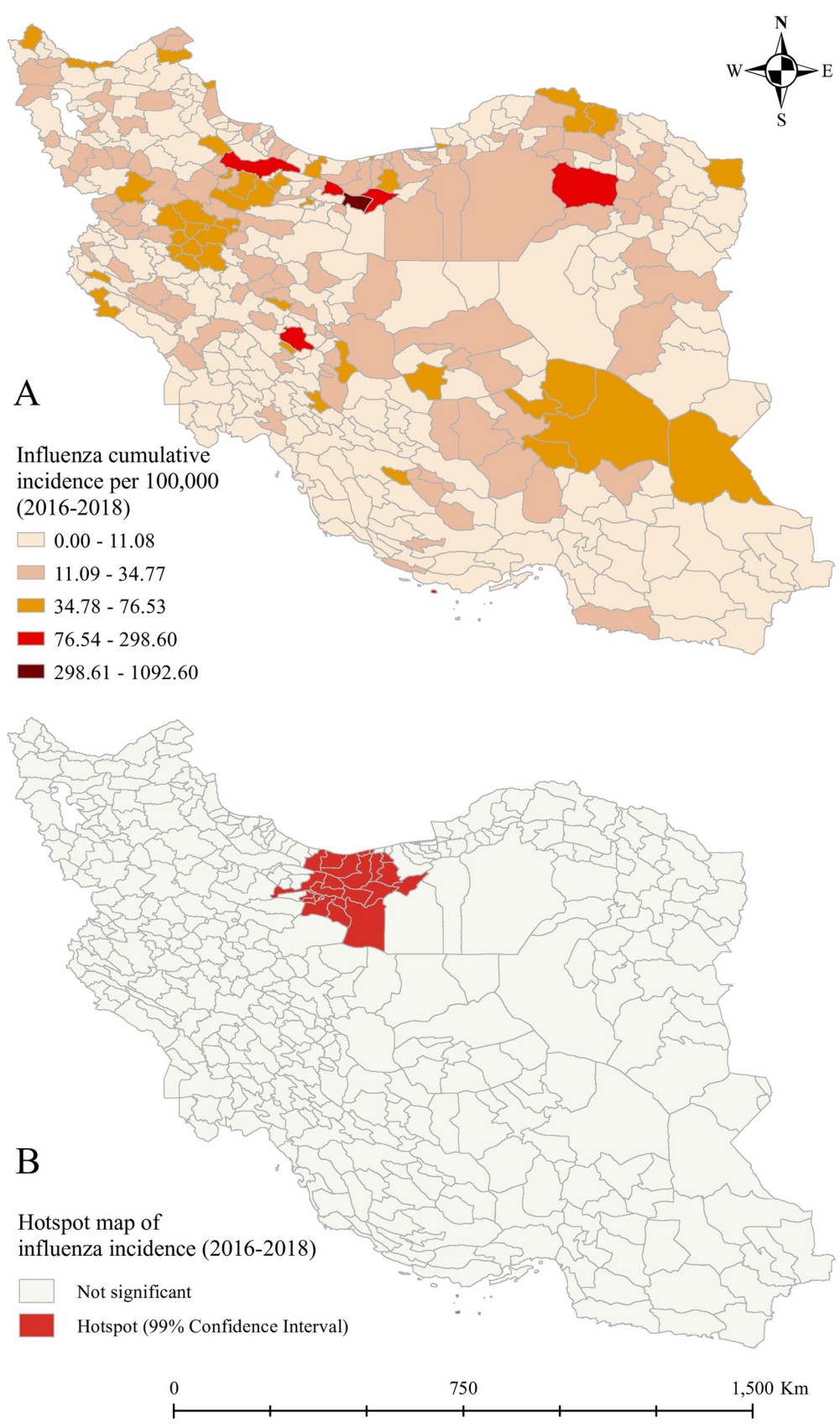

**Fig 2.** Descriptive (A) and hotspot (B) maps of total confirmed influenza cases at the county level, 2016–2018. The figure was created by authors using ArcGIS software version 10.8.

northwest regions of the country. Moreover, the increase in the cumulative incidence of influenza was attributed to late autumn to early spring, so that in the colder months (December to April), HH clusters were more developed. Women represented a higher share of hospitalized patients, whereas men had a higher share of influenza-related deaths. Older age, male sex, type-A virus and pre-existing conditions were significantly associated with higher odds of death.

Hotspot regions and HH clusters were observed in the northern parts of Iran, including the most populous counties in Iran and their neighbors in almost half of the year. In addition to

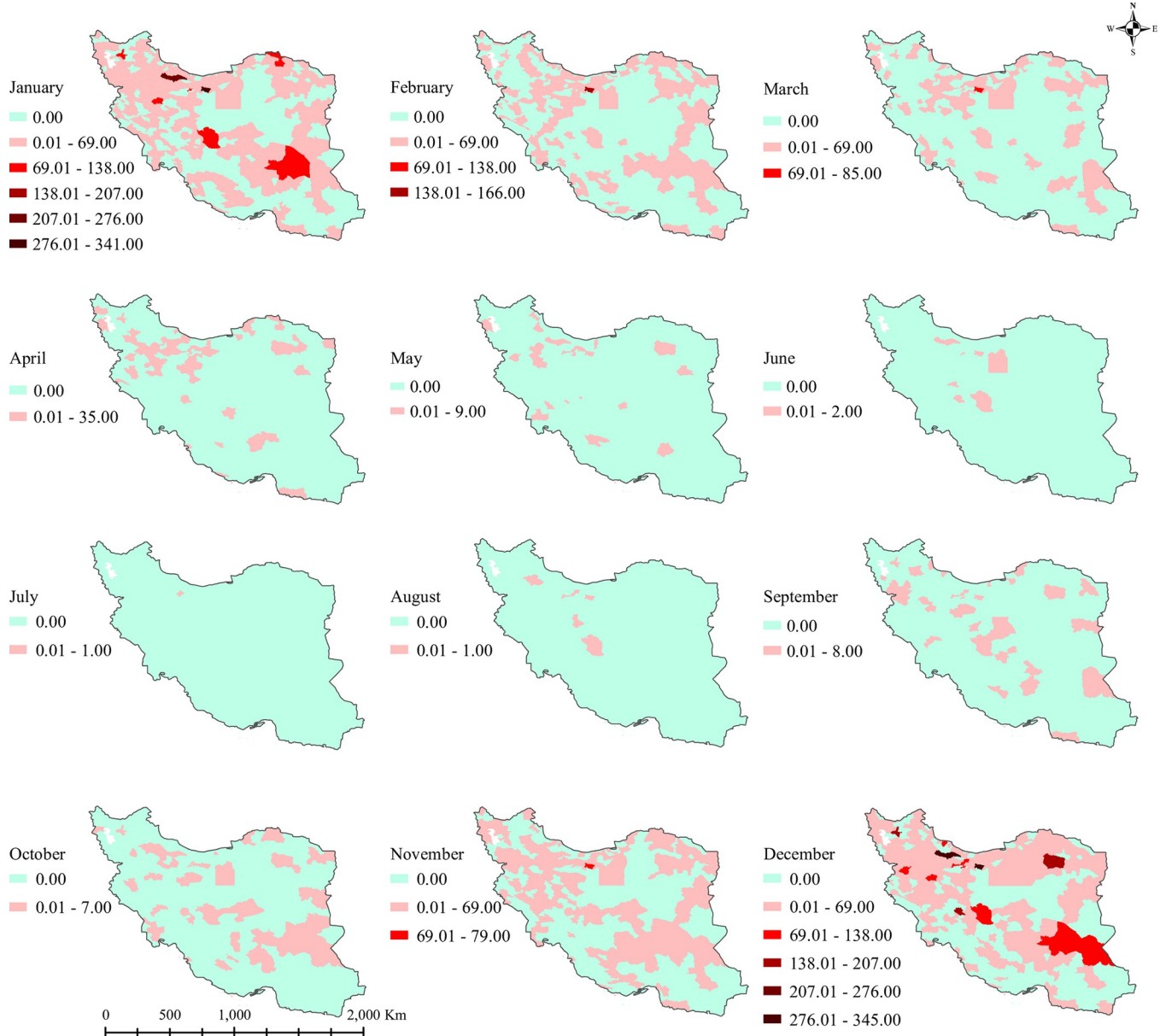

**Fig 3. County-based cumulative influenza incidence in each month, 2016–2018.** The figure was created by authors using ArcGIS software version 10.8.

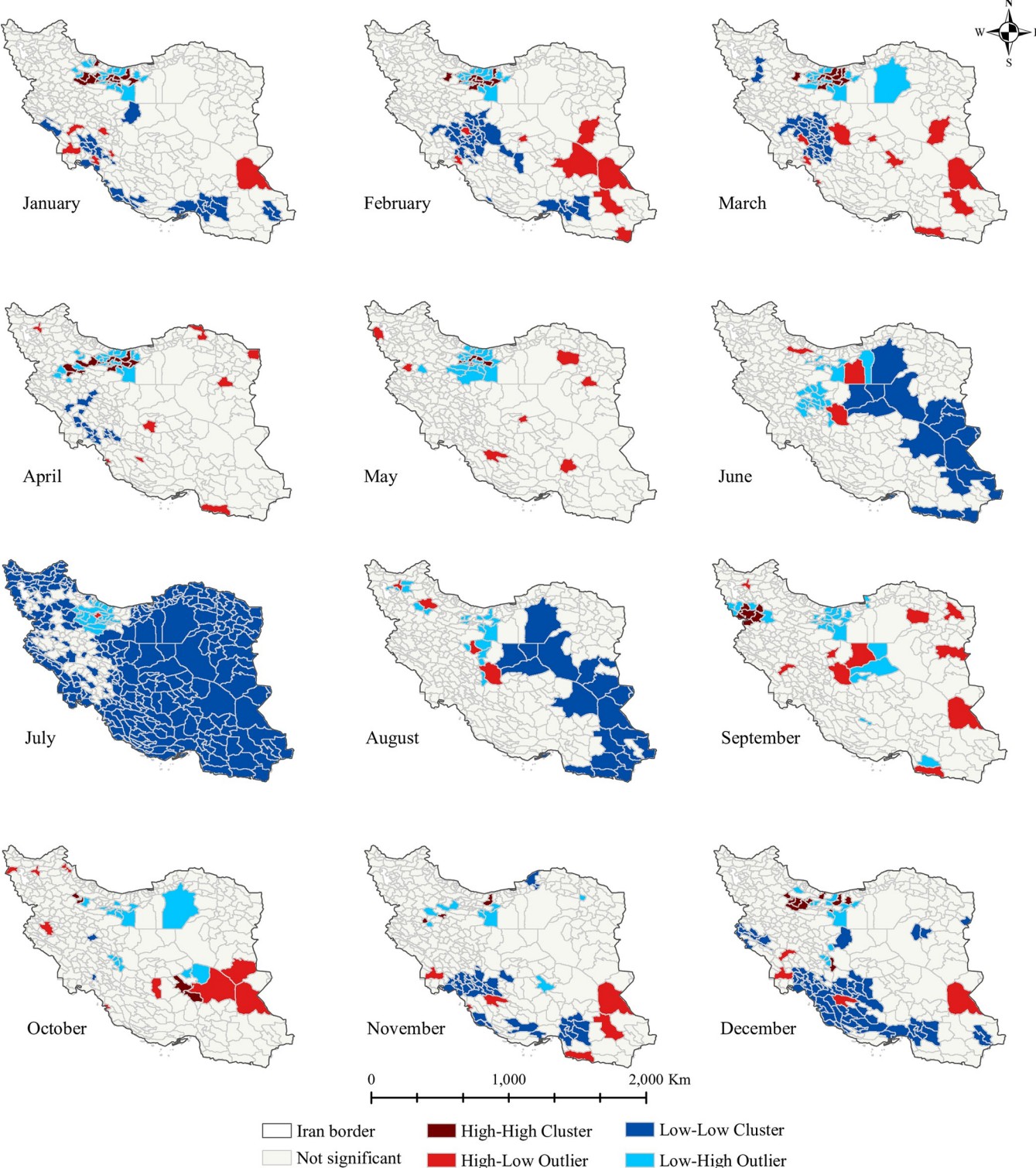

**Fig 4. Influenza clustering maps per month at the county level, 2016–2018.** High-high tracts are surrounded by tracts with high median housing values and their values are significantly higher than their neighbors, and the inverse is true for the low-low. For the tract that has a significant high-low (low-high) relationship with its neighbors, median housing values are high (low) in this tract and it is surrounded by tracts with significantly lower (higher) housing values. The figure was created by authors using ArcGIS software version 10.8.

high population density which increases their vulnerability to highly contagious pandemics [25], these areas experience higher levels of travel and commute arising from their proximity to the capital as well as tourist attractions in coastal areas of the Caspian Sea. Tehran is the most populous city in Western Asia with a population of around 9 million people in the city and 16 million across the Greater Tehran Area [26]. In such geographical areas, the spread of respiratory viruses is faster due to the further use of public transportation systems and increased network connectivity [27, 28]. Meanwhile, in big cities, the spatial heterogeneity of pollutants and their inevitable adverse effects on respiratory function can act as an important confounding factor for worsening disease outcomes and subsequently can predict a poor prognosis for influenza patients who live in congested and polluted areas [29, 30]. Moreover, given a comprehensive surveillance system and better medical care in large urban settings, the increased incidence of influenza in these areas may be a function of better screening and detection rates (i.e., surveillance bias). Therefore, big cities with advanced screening facilities have more chances to identify patients and thus become hotspots. On the opposite point, it is expected due to the lack of laboratory facilities and specialists in remote areas, the use of PCR tests is less common, and as a result, the screening and identification of patients will be limited. To reduce the burden of disease in urban areas with high transmission, it is critical to increase uptake of immunization in densely populated areas and also improve indoor and outdoor air quality.

The spatial distribution of influenza incidence was higher in the colder months (November to March) and most incident infections were observed in December and January. This is consistent with previous studies of the seasonality behavior of influenza viruses which increase the cases in winter or early spring [14, 31]. Besides, the mountainous areas, located in the vicinity of the Zagros and Alborz mountains, were affected more by influenza epidemics in the cold seasons; a finding that is compatible with previous studies showing that lower temperatures could be positively correlated with the number of influenza cases and facilitate survival and transmission of respiratory pathogens [32, 33]. Socio-economic deprivation and limited access to healthcare services in some of those remote areas, can also exacerbate the situation [34]. Additionally, the emergence of influenza in cold seasons might be aerosol-related. Aerosolization of virus-laden aerosols through coughing and sneezing in indoor dry air (relative humidity (RH) <40%), could protect the virus from the degradative effects of desiccation [35, 36]. Low RH, increased density, and more human contact in indoor spaces during the winter time could serve as a conduit for the rapid virus spread [37]. Improved indoor air quality (e.g. using humidifiers) and the avoidance of congregations in indoor spaces in cold months could probably play an important role in reducing the number of influenza cases.

Our study further identified risk factors for influenza-related mortality among hospitalized cases in Iran. Male sex was associated with increased odds of death due to influenza infection. While most previous findings in different parts of the world showed that females were more likely to die from influenza, our findings are in line with a narrow body of evidence [38–40]. Sex differences have been illustrated in antiviral treatments, engagement rates in screening and vaccination, and outcomes of influenza and could be attributed to differences in the physiological body responses and dissimilarity in environmental factors' exposures [41–43]. Children in the age range 0–9 years experienced higher infection than 10–49, which could be related to their large social networks and low practice of precautionary measures [32, 44, 45]. Furthermore, we found that older age was significantly associated with increased odds of death. This is consistent with elderly being susceptible to respiratory diseases due to a high burden of comorbidities and potential immune hypo-reactivity [6, 46]. It could also be due to low rates of immunization against influenza in Iran, especially in higher risk age groups [47, 48]. All investigated comorbidities, malignancy and grade-II obesity which are both well-established mortality risk factors in respiratory diseases, in particular [49, 50], were significantly

more common among people who died because of their infections. Over two-thirds of positive laboratory results showed type-A virus infections, more than half of which were infected solely by the H3N2 subtype. Moreover, >80% of deaths occurred among those infected with type-A which is in line with other studies conducted in the USA and across Europe [51, 52] and differs substantially from the mortality pattern of China in 2010–2012 [53]. The difference caused by the incidence and mortality rates in the present study with global burden of disease (GBD) estimations in 2017 [6] can be due to the fact that included cases considered to be influenza in GBD study, had Lower Respiratory Tract Infectious (LRTI) that they were defined as clinician-diagnosed pneumonia or bronchiolitis which their influenza virus had not been detected by reverse transcriptase (RT) PCR test. In this regard, our study included the registered data belong to hospitalized patients with positive PCR assay, which the difference in numbers and ratios can be justifiable. Identifying high-risk areas and prioritizing vaccinations for the elderly and those with severe underlying diseases can greatly reduce the irreversible consequences of influenza, including mortality. Frequent surveillance monitoring and tailored prevention programs in these areas (hot spots) and subpopulations could help reduce the burden of future influenza-related morbidity and mortality in Iran.

We acknowledge the limitations of our study. The data were based on lab-confirmed influenza patients hospitalized in medical centers and cannot be generalized to all influenza cases in Iran. Therefore, we acknowledge that the results of this study are likely non-representative of the entire population. However, given that most influenza infections are mild, we think our findings based on large national population-level hospitalizations are of particular importance for informing surveillance and controlling efforts of severe influenza cases in Iran. Nonetheless, our study is no exception to the biased nature of survival data, and we acknowledge this limitation. According to the protocols and health policies of the universities, depending on the time and the criticality of the conditions (seasonal changes or disease outbreaks such as pre-pandemic influenza conditions), the number of PCR tests in different cities could fluctuate, which is beyond the control of the authors and could affect the results. The effect of optimized surveillance systems in big cities on the detection of more cases can be one of the factors influencing the hotspots development, however, as our analyses were conducted at the county level, this effect is minor. Moreover, our dataset did not have information on the history of influenza vaccination or smoking, which would likely represent important covariates in our mortality risk factor analysis. Lastly, the issue of the modifiable areal unit problem remains inherent to the studies that focus on aggregated spatial data [54].

## Conclusions

We characterized the spatial and epidemiological features of influenza in Iran. Having comorbidities, older age, male sex, and type-A virus were associated with a worse prognosis secondary to influenza infection. Living in high-elevated and/or densely populated areas also appeared to be correlated with accelerated spread of the influenza infections. The identified high-risk areas and sub-populations could inform prioritization and geographic specificity of influenza prevention, testing, and mitigation resource management, including vaccination planning. Therefore, targeted influenza programs and preventative interventions, such as immunization, screening and treatment are warranted.

## Supporting information

**S1 Appendix. Descriptive rates and counts of Influenza infection at the provincial level, 2016–2018.**
(DOCX)

**S2 Appendix. Minimum anonymized dataset.**
(XLSX)

## Acknowledgments

The authors would like to acknowledge the assistance of the Ministry of Health and Medical Education's Center for Communicable Disease Control (CDC) and all the clinicians involved in reporting infectious cases of influenza, without whose support this research would not have been possible. The authors also acknowledge Mashhad University of Medical Sciences (MUMS) for funding this study. MK is supported by a Banting Postdoctoral Fellowship.

## Author Contributions

**Conceptualization:** Behzad Kiani, Mahnaz Arian.

**Data curation:** Zahra Rahmatinejad, Fatemeh Kiani.

**Formal analysis:** Shahab MohammadEbrahimi, Behzad Kiani.

**Funding acquisition:** Behzad Kiani.

**Investigation:** Zahra Rahmatinejad, Mohammad Dehghan-Tezerjani, Elahe Zare.

**Methodology:** Shahab MohammadEbrahimi, Behzad Kiani, Mohammad Karamouzian.

**Resources:** Mahnaz Arian, Mohammad Mehdi Gouya, Mohammad Nasr Dadras.

**Supervision:** Behzad Kiani, Stefan Baral, Mohammad Karamouzian.

**Visualization:** Shahab MohammadEbrahimi, Behzad Kiani.

**Writing – original draft:** Shahab MohammadEbrahimi.

**Writing – review & editing:** Shahab MohammadEbrahimi, Behzad Kiani, Stefan Baral, Soheil Hashtarkhani, Mohammad Karamouzian.

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
