## [Decision Letter · Decision Letter 0]

18 May 2022

PONE-D-22-08786Geospatial Epidemiology of Hospitalized Patients with a Positive Influenza Assay: A Nationwide Study in Iran, 2016-2018PLOS ONE

Dear Dr. Kiani

Thank you for submitting your manuscript to PLOS ONE. After careful consideration, we feel that it has merit but does not fully meet PLOS ONE’s publication criteria as it currently stands. Therefore, we invite you to submit a revised version of the manuscript that addresses the points raised during the review process.

We look forward to receiving your revised manuscript.

Kind regards,

Hamid Sharifi

Academic Editor

PLOS ONE

Journal Requirements:

5. We note that Figures 1, 3, 4, 5 and Supplementary File 2 in your submission contain [map/satellite] images which may be copyrighted. All PLOS content is published under the Creative Commons Attribution License (CC BY 4.0), which means that the manuscript, images, and Supporting Information files will be freely available online, and any third party is permitted to access, download, copy, distribute, and use these materials in any way, even commercially, with proper attribution. For these reasons, we cannot publish previously copyrighted maps or satellite images created using proprietary data, such as Google software (Google Maps, Street View, and Earth). For more information, see our copyright guidelines: http://journals.plos.org/plosone/s/licenses-and-copyright.

a) You may seek permission from the original copyright holder of Figures 1, 3, 4, 5 and Supplementary File 2 to publish the content specifically under the CC BY 4.0 license.  

6. Please include your tables as part of your main manuscript and remove the individual files. Please note that supplementary tables should remain as separate "supporting information" files.

Additional Editor Comments:

Dear Dr. Kiani,

Thanks so mcuh for submitting your work to PLOS ONE.

I am sharing the reviewers' comments. The main issue on this work could be related to this fact you used the surveillance data on this manuscript. The surveillance data are severly underreported. You should justidy this underreporting could not affect the results of the work.

Hamid Sharifi

Reviewers' comments:

Reviewer's Responses to Questions

**Comments to the Author**

1. Is the manuscript technically sound, and do the data support the conclusions?

Reviewer #1: Partly

Reviewer #2: Yes

Reviewer #3: Partly

2. Has the statistical analysis been performed appropriately and rigorously? 

Reviewer #1: No

Reviewer #2: Yes

Reviewer #3: N/A

3. Have the authors made all data underlying the findings in their manuscript fully available?

Reviewer #1: Yes

Reviewer #2: Yes

Reviewer #3: No

4. Is the manuscript presented in an intelligible fashion and written in standard English?

Reviewer #1: Yes

Reviewer #2: Yes

Reviewer #3: Yes

5. Review Comments to the Author

Reviewer #1: 1. Did you just use these statistical tests in the analysis? Why? (The continuous measures were compared using the Mann-Whitney U test, while dichotomous data were compared using χ2 or Fisher’s exact test, as appropriate)

2. Why did you include only “Variables with a p<0.05 in the univariable model were entered into the multivariable model”?

3. In Figure 1, you have summarized the data of two years and presented it in the figure. Due to the seasonal nature of influenza and the nature of this respiratory disease, this figure does not convey accurate information. For example, what is the reason for the high difference between provinces number 1 and number 25, which are next to each other?

4. Given the high underreporting in the influenza surveillance system, Figure 1 could probably indicate only the quality of the surveillance system or the quality diagnosis of the disease.

5. The total number of cases of the disease in the figure 2 is very small for a country with 80 million. Surveillance system seems to have a lot of underreporting. These findings can be very biased.

6. Figure 3 A may be appropriate for an article in Iran, but it does not provide more information for an international article more than Figure 1.

7. Why figure 3 B is not match with figure 1?

8. Most findings related to season, age, and risk factors for death are clear findings in the literature for influenza. The findings of the article are not novel.

9. The article discusses general points based on findings that can be misleading because they are not based on accurate data in the discus section.

10. The conclusion is very general. There is no consensus in the world on universal influenza vaccination.

11. Swine flu became an endemic disease worldwide in less than a year. The mean is, most people in the world became infected. Influenza and other viral respiratory diseases are not as preventable as mentioned in this article. Identifying hotspots is even more important to identify new strains and variants.

12. Influenza surveillance systems are extremely underreported worldwide. Therefore, instead of reporting reported cases of influenza, modeling is used for estimates. This article provides just a brief description of hospital-acquired cases of influenza

Reviewer #2: I appreciate the opportunity to review the manuscript by Kiani et al. titled “Geospatial Epidemiology of Hospitalized Patients with a Positive Influenza Assay: A Nationwide Study in Iran, 2016-2018”. The main aim of this study was to investigate the epidemiological characteristics and spatial patterns of hospitalized influenza cases in Iran. The manuscript does not contain line’s number. I will not allow this to influence my review. Detecting influenza hotspot clusters could inform policy makers, resource management, and vaccination planning to avoid potential future pandemics. Overall, the study is well-designed and well-written. There are several aspects of the study which need more clarification. Please see my comments as below:

Abstract:

The abstract is informative an contains all relevant information.

Introduction:

Introduction concisely states the background, knowledge gap and the aim of the study. I would suggest the authors to address the comments as described below:

I would ask the authors to rewrite the first sentence of the introduction “Globally, out of an annual estimated one billion cases of influenza, up to five million are severe, and as many as 650,000 infections lead to death”.

Line 6: Please change the influenza burden to “The socio-economic burden of influenza”.

Line 7: It is mentioned that cumulative incidence and mortality for influenza disease in this study were estimated at 11.44 and 0.49 (per 100,000), respectively, and case fatality rate was estimated at 4.35%. However, in introduction section (based on reference #6) it is stated that in 2017, the burden of influenza in Iran had an incidence and mortality rate of 587/100,000 people and 0.8/100,000 people, respectively.

How the authors can explain the differences in variable reported, especially in mortality?

Line 14 introduction section… “Space and time factors are among the most...” Please change space to climate.

Line 23, Introduction section “Identifying high-risk clusters of influenza could help inform Iran’s preventive and therapeutic measures.”. Please remove inform from this sentence.

Method:

Line 1, method please change this line to “Iran is located in the northeast of Persian Gulf.”

Results:

Line 1 of results section: Please state the period of cumulative incidence and also add this information to figure 1.

Demographic characteristics:

The authors mentioned that in total 9146 hospitalized influenza cases in three years were analysed. I can see the number of confirmed hospitalized cases per population are extremely low compared to those in other countries. How can the authors can explain this difference? This needs to be clarify in discussion section and also how this can affect the values for variables reported, including incidence and mortality?

Discussion:

Line 10 “Tehran is the most populous city in Western Asia with a population of around 9 million in the city and 16 million in the Greater Tehran Area”. Please include a reference for this sentence.

Line 12: “Meanwhile, spatial heterogeneity of pollutants and the negative effects of air pollution can act as a carrier of the virus and increase its spread”. This doesn’t make sense to me. Can the author explain why is it so? I was looking in reference #29 and could not find this information.

Line 14:

“Moreover, given a comprehensive surveillance system and better medical care in large urban settings, the increased incidence of influenza in these areas may be a function of better screening and detection rates”. So, this means in towns and rural areas the number of influenza cases is higher than what included in this study. How can this affect the spatial distribution and hotspot maps of influenza in Iran?

Line 15: “To reduce the burden of disease in these areas there is a need to control and reduce air pollution in large cities..” I would suggest the authors discuss how air pollution can increase the number of influenza cases and cite some appropriate references.

Study limitations and strengths need to be added to the discussion section.

Conclusion:

Please add more information to this section. The major findings of your study need to be added into this section.

Reviewer #3: I read the manuscript and two important comments which are as follows:

1. In the method section, it has been written that multivariate logistic regression has been used for analysis. As you know, how variables are coded plays a very important role in interpreting the results of this regression. But in this methodology, how to encode any of the variables entered into the regression model has not been stated.

2. In the method section, it has been written the data of confirmed cases (i.e. positive PCR) has been used to perform calculations and draw a map of the spatial distribution of influenza. The first point is that in the executive field, performing PCR diagnostic test for influenza cases is not common and this test is performed only at the request of the patient’s treating specialist in certain circumstances (for example, the person is likely to be hospitalized in the ICU and to achieve a definitive diagnosis and to avoid infecting other critically ill patients admitted to the ICU on a case-test basis, PCR test is requested), which also this is not a routine measure of influenza diagnosis in most hospitals in the country, and the “clinical diagnosis” of a physician is the criterion for hospitalization. The second point is that a significant number of possible hospitalized cases of influenza are eventually coded under the heading of severe respiratory syndrome, respiratory disorder, or pneumonia according to “ICD-10” at the time of discharge, and their coding is not precisely and specifically done for influenza, and therefore a significant number of patients with definitive influenza that forms a part of data, is not reported in this article due to registration in another code (for example severe respiratory syndrome, respiratory disorder, pneumonia). The third point is that around 2018, for example, due to numerous “clinical diagnosis” reports of influenza and flu-like illness cases and the decision of the medical system to confirm the diagnosis and rapid response to the outbreak, some provincial medical universities (such as Gorgan, and East Azerbaijan provinces) were allowed to perform PCR tests to diagnose suspected hospitalizations. Therefore, the number of tests and, consequently, the number of positive cases identified due to these tests increased, but similar tests has not been done in some of the Iranian provinces or has not been given much attention which might indeed lead to unrepresentative data.

I believe that, all of the above-mentioned items cause that the estimates and maps obtained from this data do not reflect the real picture of the problem, and also is prone to significant underestimation. For non-professional readers who are not familiar with conventional clinical processes, may be lead to misrepresenting high-risk and low-risk areas.

6. PLOS authors have the option to publish the peer review history of their article (what does this mean?). If published, this will include your full peer review and any attached files.

Reviewer #1: No

Reviewer #2: No

Reviewer #3: No

---

## [Author Response · Author response to Decision Letter 0]

19 Oct 2022

Journal Requirements:

1. Please ensure that your manuscript meets PLOS ONE's style requirements.

Response: All the style requirements were met following the PLOS ONE’s style templates.

2. Please provide additional details regarding participant consent.

Response: In this study, all data were fully anonymized. Therefore, the ethical committee waived the need for informed consent from individual patients. In the revision step, this statement has been added to the article’s Methods section > Ethics Statement.

3. Regarding data availability, if there are no restrictions, please upload the minimal anonymized data set necessary to replicate your study findings.

Response: Thank you for your suggestion. We have uploaded all the minimum anonymized dataset via S2 Appendix.

4. Your ethics statement should only appear in the Methods section of your manuscript.

Response: Thank you. We have revised our manuscript and we confirm that this section has only appeared in the Methods section of the manuscript (Methods section > Ethics Statement). 

5. Figures 1, 3, 4, 5 and Supplementary File 2 in your submission contain [map/satellite] images which may be copyrighted. Please check copyright information on all replacement figures and update the figure caption with source information.

Response: Thanks for your note. As we have mentioned in Methods > Software section, all the descriptive maps and spatial analyses in this study were created by the authors using ArcGIS software, version 10.8. Therefore, we did not need to obtain any permission or license to publish Figures 1, 3, 4, 5 and Supplementary File 2. The figures do not include any data from any third-party software or company. Based on this comment and to clarify this for the readers, we have added the following sentence to the captions of Figures 2, 3, 4. 

“The figure was created by authors using ArcGIS software version 10.8.”

Please note that according to the reviewers’ comments, we have also decided to remove Figure 1 and Supplementary file 2 (Descriptive and cluster maps of influenza at the provincial level) from the previous version in the revision step. We have updated all figure citations, captions, and file names accordingly.

6. Please include your tables as part of your main manuscript and remove the individual files. Please note that supplementary tables should remain as separate "supporting information" files.

Response: Thank you. We have included Tables 1 and 2 inside the manuscript file and kept the supplementary table as S1 Appendix.

7. Please include captions for your Supporting Information files at the end of your manuscript, and update any in-text citations to match accordingly.

Response: Supplementary file 2 was removed in the revision process, however, a new Supplementary file (S2 Appendix) has been added to the manuscript according to your comments regarding uploading the raw data. For both Supporting Information files, captions have been added properly at the end of manuscript as S1 & S2 Appendix. The captions are represented as follows:

“S1 Appendix Influenza incidence and fatality rates at the provincial level, 2016-2018”

“S2 Appendix Minimum anonymized dataset”

Editor Comment:

I am sharing the reviewers' comments. The main issue on this work could be related to this fact you used the surveillance data on this manuscript. The surveillance data are severely underreported. You should justify this underreporting could not affect the results of the work.

Response: Dear editor, thank you for your attention and for interest. We agree that the problem raised by the reviewers could be seen as a significant concern and should be more extensively discussed. As mentioned in the manuscript, we used the data of hospitalized Influenza patients over three years, all of whom tested positive for PCR. In Iran, the Influenza surveillance system was established since 2004 by the Ministry of Health to improve the involvement of medical associations in reporting Influenza and enhance data sharing with the Ministry of Health (2). After that, the PCR test was further required for patients suspected of having Influenza who needed to be hospitalized. Therefore, people who are suffering from influenza and require hospitalization must take a PCR test for two main reasons; A) to avoid infecting other hospitalized patients, and B) to report the exact number of confirmed Influenza cases. This information must be periodically reported from the vice-chancellor of the provincial medical universities to the Ministry of Health. Therefore, our study sample did not include people who had a common cold, Influenza-like illness (ILI), clinical diagnosis of influenza, or those who were treated at home. Therefore, as clearly stated in the manuscript and the title, our data is not trying to represent all Influenza cases in Iran and is only limited to those described above. We have also discussed this important issue in the Discussion of the paper (Discussion > Fourth paragraph & Limitation section).

Reviewers' comments:

Reviewer #1: 

1. Did you just use these statistical tests in the analysis? Why? (The continuous measures were compared using the Mann-Whitney U test, while dichotomous data were compared using χ2 or Fisher’s exact test, as appropriate)

Response: Thank you for your question and sorry for the confusion. According to the characteristics and distribution of the data, we used these statistics for primary statistical analyses. However, as described in the Methods, this study has also used more advanced statistical analyses such as multivariable regression and spatiotemporal analysis. We would like to kindly draw your attention that the most important aspect of our investigations was related to spatiotemporal context. Therefore, we did not only use the basic statistical analysis mentioned in this section. Based on your comment, we decided to rename the titles of our Method’s sub-sections to show that the spatial analysis conducted in this study is also a kind of statistical analyses (Methods > Data Analysis > Statistical Analysis). Therefore, in the current version of the manuscript, the statistical analysis section includes sub-sections of ‘Descriptive statistics and multivariable regression’, ‘Cluster and outlier analysis’, and ‘Hotspot analysis’. We hope this addresses the confusion around this issue.

2. Why did you include only “Variables with a p<0.05 in the univariable model were entered into the multivariable model”?

Response: Thank you for your note. We used the multivariable logistic regression to provide a more objective approach for studying the effects of covariates (such as age, sex, comorbidities etc.) on the binary outcome (Death/ Survive). The logistic regression has been approved as one of the correct statistical models for this research question (3). We entered variables with p<0.05 in the univariable model into the multivariable logistic regression model, which is one of the acceptable approaches that can be used to choose what variables should be included in regression analysis (4). This filtering system helps to avoid adding extra variables in the logistic regression, which can cause an unrealistic model. Although the value of 0.20 for alpha level is often used for screening variables, because of the number of variables and to provide better predictions in our big dataset (5), we used 0.05 as the threshold in this study to be on the conservative side. Indeed, given that the P-values in the univariable analysis are all highly significant (Lowest is 0.012), choosing a higher p-value cutoff (e.g., 0.15 or 0.2) would have not led to any differences in the variables that were entered into the multivariable model. Such a univariable analysis screening (UAS) method to select covariates for multivariable logistic regression has been widely used in research studies published in previous literature (4,6). According to this comment, we have revised this section and added some sentences for more clarification (Methods > Data Analysis > Statistical Analysis > Descriptive statistics and multivariable regression).

3. In Figure 1, you have summarized the data of two years and presented it in the figure. Due to the seasonal nature of influenza and the nature of this respiratory disease, this figure does not convey accurate information. For example, what is the reason for the high difference between provinces number 1 and number 25, which are next to each other?

Response: Thank you for your question and sorry for the confusion. We had used Figure 1 to show the study area. However, based on your comment and because our analyses were conducted at the county level while this figure has provided the rates at the provincial level, we have decided to remove this figure from our manuscript. We have updated all figure citations, captions, and file names accordingly.

4. Given the high underreporting in the influenza surveillance system, Figure 1 could probably indicate only the quality of the surveillance system or the quality diagnosis of the disease.

Response: As mentioned in the Methods section, our data included all inpatient cases hospitalized due to influenza in this period, confirmed by a positive laboratory test using the real-time reverse transcription-polymerase chain reaction (RT-PCR) assay (n=9146). Therefore, we have lost many Influenza cases that were treated at home or referred to a healthcare center for follow-up treatment without hospitalization or any PCR test. These kinds of patients represent the ‘Influenza-Like Illness (ILI)’. However, we have mentioned this issue in the title and the other appropriate places such as the fourth paragraph of the Discussion. Of course, it should be mentioned that according to your previous comment and due to the provincial scale of the analyses, Figure 1 has been removed from the manuscript.

5. The total number of cases of the disease in the figure 2 is very small for a country with 80 million. Surveillance system seems to have a lot of underreporting. These findings can be very biased.

Response: Thank you for your note. We have double-checked these statistics and confirm that they are accurate. As noted earlier, the small number of cases for a country of 80 million people is because these numbers ONLY refer to hospitalized cases with a confirmed PCR test and do not include all influenza cases. 

Please also note that only half a percent of all influenza cases experiences severe condition and need hospitalization (1). Therefore, our findings are not biased. We have made sure these limitations are clearly laid out in the Limitations. 

6. Figure 3 A may be appropriate for an article in Iran, but it does not provide more information for an international article more than Figure 1.

Response: Thank you for pointing this out. As you correctly mentioned, Figure 1 and this figure are closely related, showing cumulative incidence rates. The difference is that the former is based on the provincial division, and the latter is based on the counties division. According to this comment and your former comments, we decided to remove Figure 1 because it was at the provincial level. Therefore, we have kept Figure 3 A (now it has been captioned as Figure 2 A) in the manuscript.

7. Why figure 3 B is not match with figure 1?

Response: Figure 3 B (Figure 2 B in revised version) contains an Iran map that shows some hot spot counties. This map is developed based on the Getis-Ord Gi* analysis, a hot spot analysis method that looks at each feature in the dataset within the context of neighboring features in the same dataset. There may be a feature with a high value in Figure 1, but it may not be a statistically significant hotspot. In order to be a significant hotspot, a feature with a high value will be surrounded by other features with high values. In other words: “The local sum for a feature and its neighbors is compared proportionally to the sum of all features; when the local sum is very different from the expected local sum, and that difference is too large to be the result of random choice, a statistically significant z-score results” (7). Accordingly, Figure 3 B shows some northern counties with high Influenza cases surrounded by other high values counties. But Figure 1 tells a different story and those values indicate the cumulative incidence of influenza cases in each province. According to this comment, we have added more details in the manuscript clarifying what hotspots are (Methods > Data Analysis > Statistical Analysis > Hotspot analysis). Also, as mentioned before, we removed Figure 1. 

8. Most findings related to season, age, and risk factors for death are clear findings in the literature for influenza. The findings of the article are not novel.

Response: We agree that descriptive reports of Influenza patients have rich literature, and this study does not add a novel result to it, but as we explained in previous comments, the focus of this paper is on hospitalized patients with a positive PCR test. However, in developing countries such as Iran, resource limitation is a big problem, and because of that, presenting research-based knowledge can be very helpful for health policy-makers to take into account their tailored actions. This study has covered two major knowledge gaps, a) What are the epidemiological characteristics of hospitalized Influenza patients, and b) What are the hidden spatial patterns (hot spots/ clustering) of Influenza incidence across Iran. The spatial findings of this study are worthwhile. These can be used in national macro-decisions to monitor and control infectious diseases. Moreover, other countries in the Eastern Mediterranean region could use this methodological approach to research their needs appropriately. Finally, it should be noted that to the best of our knowledge, such a study (a mixture of epidemiological and spatial analysis with an emphasis on discovering the hidden geographical patterns of disease occurrence) has not recently been done on PCR-confirmed Influenza cases in Iran. 

9. The article discusses general points based on findings that can be misleading because they are not based on accurate data in the discus section.

Response: Thanks for your comment. As we responded in the previous comments, this study focused on the hospitalized patients that their influenza has been confirmed by the PCR test. So, it is true that our results will be different from the literature of influenza research without PCR confirmation of Influenza in the general population, but might not be misleading because it is clear to readers that our dataset is different and unique. 

10. The conclusion is very general. There is no consensus in the world on universal influenza vaccination.

Response: Yes, you are definitely right, and we agree that there is no consensus on influenza immunization. We are sorry for the confusion but we are not advocating for influenza vaccination for everyone. Considering the resource-limited nature of Iran, GIS can be a helpful tool to identify and prioritize populations so that people living in high-risk places would receive these types of services faster. These services could include vaccination, screening, and treatment. Based on this comment, the Conclusion has been updated. 

11. Swine flu became an endemic disease worldwide in less than a year. The mean is, most people in the world became infected. Influenza and other viral respiratory diseases are not as preventable as mentioned in this article. Identifying hotspots is even more important to identify new strains and variants.

Response: Thank you for pointing this issue out. As you correctly mentioned, identifying hot spots for disease occurrence is highly valuable. However, it is not a perfect tool to 'prevent' the diseases from spreading. Health policymakers will be aware of what regions are more susceptible to the high incidence of the disease, where and when they can take proper action to reduce irreparable damage to the community, and what the next action step is. Hence, “disease monitoring and control using GIS” is better than “disease prevention”. The manuscript has been checked for this, and corrections have been made where necessary.

12. Influenza surveillance systems are extremely underreported worldwide. Therefore, instead of reporting reported cases of influenza, modeling is used for estimates. This article provides just a brief description of hospital-acquired cases of influenza.

Response: We agree that surveillance data might suffer from underreporting. However, as you mentioned, our study is based on hospitalized influenza cases, which might have better quality as it has a medical registry in Iran. Efficient and reliable surveillance data are vital for monitoring public health trends and disease outbreaks. However, there are limitations associated with the use of data from surveillance and notification systems since they are affected by a degree of uncertainty which is unavoidable (especially for infectious diseases). This uncertainty can occur on two separate levels; a) at the community-level and b) at the healthcare-level. In this regard, Gibbons et al. clarified based on a ‘Morbidity Surveillance Pyramid’ that how many percent of real infected cases will report on average, which is only about 29% of all cases (8), which undoubtedly, in developing countries like Iran, this percentage will be lower. The Influenza pathogen type was identified in the biochemical laboratories using PCR-RT test for all the considered patients. Therefore, we can be sure that all these patients are definite and confirmed cases that have suffered from acute conditions due to influenza. As another point, in infectious epidemics, short-term alterations in surveillance data investigation may affect more on the interpretation of disease trends (9), which we have covered long-term alternations of the study period (three years) to avoid this as much as possible. With all that said, our study is no exception to the biased nature of survival data and we acknowledge this limitation and have noted it in the Limitations section (Discussion > Last paragraph).

Reviewer #2: 

I appreciate the opportunity to review the manuscript by Kiani et al. titled “Geospatial Epidemiology of Hospitalized Patients with a Positive Influenza Assay: A Nationwide Study in Iran, 2016-2018”. The main aim of this study was to investigate the epidemiological characteristics and spatial patterns of hospitalized influenza cases in Iran. The manuscript does not contain line’s number. I will not allow this to influence my review. Detecting influenza hotspot clusters could inform policy makers, resource management, and vaccination planning to avoid potential future pandemics. Overall, the study is well-designed and well-written. There are several aspects of the study which need more clarification. Please see my comments as below:

Abstract: 

The abstract is informative and contains all relevant information.

Response: Thank you so much for your positive feedback.

Introduction: 

Introduction concisely states the background, knowledge gap and the aim of the study. I would suggest the authors to address the comments as described below:

1. I would ask the authors to rewrite the first sentence of the introduction “Globally, out of an annual estimated one billion cases of influenza, up to five million are severe, and as many as 650,000 infections lead to death”. 

Response: It has been updated.

2. Line 6: Please change the influenza burden to “The socio-economic burden of influenza”.

Response: Many thanks, it has been done.

3. Line 7: It is mentioned that cumulative incidence and mortality for influenza disease in this study were estimated at 11.44 and 0.49 (per 100,000), respectively, and case fatality rate was estimated at 4.35%. However, in introduction section (based on reference #6) it is stated that in 2017, the burden of influenza in Iran had an incidence and mortality rate of 587/100,000 people and 0.8/100,000 people, respectively. How the authors can explain the differences in variable reported, especially in mortality?

Response: Thank you for your consideration. The mentioned paper reported that the included cases considered to be influenza had Lower Respiratory Tract Infectious (LRTI) and other respiratory conditions like Chronic Obstructive Pulmonary Disease (COPD). LRTIs and COPDs were defined as clinician-diagnosed pneumonia or bronchiolitis, which their influenza virus had not been detected by reverse transcriptase (RT) PCR test. In addition, this study is a modeling and estimation study, not a detailed report of registered data. Undoubtedly, we have lost many Influenza cases that had home treatment or refer to a health center for follow-up treatment without hospitalization or any PCR test. As a result, our study included the registered data belonging to hospitalized patients with positive PCR assay; the difference in numbers and ratios can be justifiable (10). This difference was discussed in the Discussion section (Fourth paragraph).

4. Line 14 introduction section “Space and time factors are among the most...” Please change space to climate.

Response: Sorry for confusion. In fact, the meaning of space and time here is related to spatial and temporal analysis (a commonly used phrase in spatial studies). Climate is a spatial factor but does not include other geographical factors. According to your comment, we have changed it to “Spatiotemporal factors” to avoid confusion.

5. Line 23, Introduction section “Identifying high-risk clusters of influenza could help inform Iran’s preventive and therapeutic measures.” Please remove inform from this sentence.

Response: Thank you, it has been removed.

Methods:

6. Line 1, method please change this line to “Iran is located in the northeast of Persian Gulf.”

Response: Thanks a lot, it has been changed.

Results:

7. Line 1 of results section: Please a) state the period of cumulative incidence and also b) add this information to figure 1. Demographic characteristics:

Response: Thank you. a) The first paragraph of the Results has been updated as you mentioned. b) Based on the feedback from reviewer 1, we have removed Figure 1 from the manuscript and this comment no longer applies.

8. The authors mentioned that in total 9146 hospitalized influenza cases in three years were analyzed. I can see the number of confirmed hospitalized cases per population are extremely low compared to those in other countries. How the authors can explain this difference? This needs to be clarify in discussion section and also how this can affect the values for variables reported, including incidence and mortality?

Response: As it was mentioned in the Methods section, our data included all inpatient cases who had been hospitalized due to influenza in this period, confirmed by a positive laboratory test using the real-time reverse transcription-polymerase chain reaction (RT-PCR) assay (n=9146). All these data were obtained from the infectious disease registry of the Ministry of Health, which was launched in 2004, and all of them are of high quality and accuracy. However, to be sure of the total number, we have double-checked this with the Ministry of Health, the number of PCR-confirmed Influenza cases is correct and there is no problem. Therefore, the small number of cases for a country of 80 million people can be because all these patients were in severe condition and had not self-medicate, so they needed to be hospitalized and receive clinical care. Based on that, undoubtedly, we have lost many Influenza cases that had home treatment or refer to a health center for follow-up treatment without hospitalization or any PCR test. So, we can say this study only covered severe cases of the disease, and as estimated by Iuliano et al. (1), presumably this is only 0.5% of the total population of people who get the Influenza in Iran. Therefore, this number can be reasonable. We discussed the difference between our study and other research at the end of the Discussion section (Discussion > Fourth paragraph).

Discussion:

9. Line 10 “Tehran is the most populous city in Western Asia with a population of around 9 million in the city and 16 million in the Greater Tehran Area”. Please include a reference for this sentence.

Response: Thank you for your attention. We referred to a study mentioning this information. Reference No. #26 (Discussion > Second paragraph), by Effati et al. (11). 

10. Line 12: “Meanwhile, spatial heterogeneity of pollutants and the negative effects of air pollution can act as a carrier of the virus and increase its spread”. This doesn’t make sense to me. Can the author explain why it is so? I was looking in reference #29 and could not find this information.

Response: Thank you for your note. As Karimi et al. (12) concluded, all air pollutants were related to increased risk of respiratory disease and mortality, which the effects of PM2.5, PM10, SO2, and NO2 were significant. To estimate the values of a spatial variation of pollutants concentrations in Tehran, the ordinary kriging technique was used to estimate the values of a spatial variation of pollutant concentrations. Based on that, spatial heterogeneity of pollutants and air pollution has a significant association with increased respiratory diseases and mortality. However, this sentence that you mentioned has some semantic problems, therefore we have updated it to the following sentence: (Discussion > Second paragraph > Line 7)

 “Meanwhile, in big cities, the spatial heterogeneity of pollutants and their inevitable adverse effects on respiratory function can act as an important confounding factor for worsening disease outcomes and subsequently can predict a poor prognosis for influenza patients who live in congested and polluted areas” (12).

11. Line 14: “Moreover, given a comprehensive surveillance system and better medical care in large urban settings, the increased incidence of influenza in these areas may be a function of better screening and detection rates”. So, this means in towns and rural areas the number of influenza cases is higher than what included in this study. How can this affect the spatial distribution and hotspot maps of influenza in Iran?

Response: Based on the literature, we know that surveillance system has a better functionality in main cities and places that are considered as a connection point between other peripheral points. Accordingly, the screening, identification and treatment of patients (with any type of disease) will be done more and better in these urban settings than the other remote areas. It is true that the PCR test is easily accessible in many cities of Iran, but the thing to consider is that unless there is a well-equipped laboratory and an expert person to perform this test, this diagnosis will not be made. Therefore, in remote areas, due to the lack of facilities and specialists, this test will inevitably be performed less often, and as a result, the screening and identification of patients will be limited. So, it is hypothesized that the high incidence rate of Influenza in big cities like Tehran and its neighbors (such as Alborz) is not just because of high population density or air pollution, but it can be due to better screening and more efficient surveillance system, which has caused the hotspots development in these areas. We have added this to the Limitation section. This can affect the spatial distribution but because we have analyzed our data at the county level, not the city level, this effect is the least. We have enriched our statements. (Discussion > Second paragraph)

12. Line 15: “To reduce the burden of disease in these areas there is a need to control and reduce air pollution in large cities...” I would suggest the authors discuss how air pollution can increase the number of influenza cases and cite some appropriate references.

Response: As we previously mentioned this in comment #10, air pollution has a significant relationship with increasing respiratory morbidity and mortality. The pollutants can have a major effect on respiratory physiological functions and subsequently can predict a poor prognosis for Influenza patients who live in congested and polluted areas. We have discussed it in the Discussion. (Discussion > Second paragraph)

13. Study limitations and strengths need to be added to the discussion section.

Response: Thank you, it has been done.

Conclusion:

14. Please add more information to this section. The major findings of your study need to be added into this section.

Response: Thank you for pointing this out. The Conclusion has been revised as follow: 

“We characterized the spatial and epidemiological features of influenza in Iran. Having comorbidities, older age, male sex, and type-A virus were associated with a worse prognosis secondary to influenza infection. Living in high-elevated and/ or densely populated areas also appeared to be correlated with accelerated spread of the influenza infections. The identified high-risk areas and sub-populations could inform prioritization and geographic specificity of influenza prevention, testing, and mitigation resource management including vaccination planning. Therefore, targeted influenza programs and preventative interventions such as immunization, screening and treatment, are warranted.”

Reviewer #3: 

I read the manuscript and two important comments which are as follows: (In order to respond more clearly, we have separated reviewer’s comments to five sections)

1. In the method section, it has been written that multivariate logistic regression has been used for analysis. As you know, how variables are coded plays a very important role in interpreting the results of this regression. But in this methodology, how to encode any of the variables entered into the regression model has not been stated.

Response: Thank you for this valuable comment. As it has been shown in Table 2, we had two types of variables to develop a logistic regression model, continuous and categorical variables. The only continuum variable was ‘Age’ which is marked with an asterisk and this phrase is mentioned in the footnote of the table: “Treated as a continuous variable”. Other variables used in the model are binary, including virus type (A vs. B), sex (male vs. female), and comorbidities (yes vs. no). However, based on your comment, we have added ‘Reference’ for all the variables to Table 2 (Results> Table 2) for more clarity and also a clarification statement in the Methods section has been mentioned (Methods > Data Analysis > Statistical Analysis > Descriptive statistics and multivariable regression).

2. In the method section, it has been written the data of confirmed cases (i.e. positive PCR) has been used to perform calculations and draw a map of the spatial distribution of influenza. The first point is that in the executive field, performing PCR diagnostic test for influenza cases is not common and this test is performed only at the request of the patient’s treating specialist in certain circumstances (for example, the person is likely to be hospitalized in the ICU and to achieve a definitive diagnosis and to avoid infecting other critically ill patients admitted to the ICU on a case-test basis, PCR test is requested), which also this is not a routine measure of influenza diagnosis in most hospitals in the country, and the “clinical diagnosis” of a physician is the criterion for hospitalization. 

Response: Thank you for pointing this out. You are right and we know that most Influenza patients have self-treatment, or they don't even need to go to medical centers at all. Even many these patients, after visiting medical centers and receiving medicine, are advised to rest at home by their treating specialist and do not need to be hospitalized, which in this case, the “clinical diagnosis” of the attending physician is sufficient. In Iran, the Influenza surveillance system has been established since 2004 by the Ministry of Health to more the involvement of medical associations in reporting Influenza and precise data sharing with the Ministry of Health (2) . After that, the PCR test was more required for patients suspected of having Influenza who needed to be hospitalized. Therefore, most people who are suffering from influenza and need to be hospitalized are required to take a PCR test for two main reasons; A) to avoid infecting other hospitalized patients, and B) to report the exact number of Influenza cases. This information must be periodically reported from the vice-chancellor of the provincial medical universities to the Ministry of Health. Therefore, in this study, we have focused on the data of patients who were in severe conditions of respiratory infection, their PCR test was positive, and they were also admitted to the hospital and received medical services. To clarify the difference between the data of this study and other studies, some changes have been made in the manuscript (Discussion> Paragraph 4 and 5).

3. The second point is that a significant number of possible hospitalized cases of influenza are eventually coded under the heading of severe respiratory syndrome, respiratory disorder, or pneumonia according to “ICD-10” at the time of discharge, and their coding is not precisely and specifically done for influenza, and therefore a significant number of patients with definitive influenza that forms a part of data, is not reported in this article due to registration in another code (for example severe respiratory syndrome, respiratory disorder, pneumonia).

Response: Thank you for your important note. As stated in the previous comment, we did not retrieve the required data from patients' medical records. Therefore, the different coding of their diagnoses (by ICD-10) did not affect the output of our study. We only used the Influenza surveillance system data designed for the registration of infectious diseases by the Ministry of Health, and all provincial medical universities in the country must register their new infectious cases in this system, regardless of their ICD-10 codes in the patient records.

4. The third point is that around 2018, for example, due to numerous “clinical diagnosis” reports of influenza and flu-like illness cases and the decision of the medical system to confirm the diagnosis and rapid response to the outbreak, some provincial medical universities (such as Gorgan, and East Azerbaijan provinces) were allowed to perform PCR tests to diagnose suspected hospitalizations. Therefore, the number of tests and, consequently, the number of positive cases identified due to these tests increased, but similar tests has not been done in some of the Iranian provinces or has not been given much attention which might indeed lead to unrepresentative data. 

Response: Thank you for this detailed comment about the study area. A large number of tests can affect the results but why many tests have been conducted in a specific area? As you mentioned, it was due to numerous “clinical diagnoses” which means there are a lot of real influenza cases. In this case, it is not surprising if it leads to hotspots in these areas. Anyway, according to this comment, we have added this issue to the Limitation section (Discussion> Last paragraph). 

5. I believe that, all of the above-mentioned items cause that the estimates and maps obtained from this data do not reflect the real picture of the problem, and also is prone to significant underestimation. For non-professional readers who are not familiar with conventional clinical processes, may be lead to misrepresenting high-risk and low-risk areas.

Response: Thanks for this comment. We did our best in our manuscript to reflect the main difference of our data with the other available influenza data sources, mostly in the Methods and Discussion sections. We believe that this is the difference of our study, and every reader might consider our results in this setting. 

References

1. Iuliano AD, Roguski KM, Chang HH, Muscatello DJ, Palekar R, Tempia S, et al. Estimates of global seasonal influenza-associated respiratory mortality: a modelling study. The Lancet. 2018 Mar 31;391(10127):1285–300. 

2. Al Awaidi S, Abusrewil S, AbuHasan M, Akcay M, Aksakal FNB, Bashir U, et al. Influenza vaccination situation in Middle-East and North Africa countries: Report of the 7th MENA Influenza Stakeholders Network (MENA-ISN). J Infect Public Health. 2018 Nov 1;11(6):845–50. 

3. PRENTICE RL, PYKE R. Logistic disease incidence models and case-control studies. Biometrika. 1979 Dec 1;66(3):403–11. 

4. Postlewait LM, Ethun CG, McInnis MR, Merchant N, Parikh A, Idrees K, et al. Association of Preoperative Risk Factors With Malignancy in Pancreatic Mucinous Cystic Neoplasms: A Multicenter Study. JAMA Surg. 2017 Jan 1;152(1):19–25. 

5. Bursac Z, Gauss CH, Williams DK, Hosmer DW. Purposeful selection of variables in logistic regression. Source Code Biol Med. 2008 Dec 16;3(1):17. 

6. Nor AM, Davis J, Sen B, Shipsey D, Louw SJ, Dyker AG, et al. The Recognition of Stroke in the Emergency Room (ROSIER) scale: development and validation of a stroke recognition instrument. Lancet Neurol. 2005 Nov 1;4(11):727–34. 

7. clubdebambos. What is Hotspot Analysis? [Internet]. Geospatiality. 2016 [cited 2022 Jun 25]. Available from: https://glenbambrick.com/2016/01/21/what-is-hotspot-analysis/

8. Gibbons CL, Mangen MJJ, Plass D, Havelaar AH, Brooke RJ, Kramarz P, et al. Measuring underreporting and under-ascertainment in infectious disease datasets: a comparison of methods. BMC Public Health. 2014 Feb 11;14(1):147. 

9. Kovacevic A, Eggo RM, Baguelin M, Domenech de Cellès M, Opatowski L. The Impact of Cocirculating Pathogens on Severe Acute Respiratory Syndrome Coronavirus 2 (SARS-CoV-2)/Coronavirus Disease 2019 Surveillance: How Concurrent Epidemics May Introduce Bias and Decrease the Observed SARS-CoV-2 Percentage Positivity. J Infect Dis. 2022 Jan 18;225(2):199–207. 

10. Troeger CE, Blacker BF, Khalil IA, Zimsen SRM, Albertson SB, Abate D, et al. Mortality, morbidity, and hospitalisations due to influenza lower respiratory tract infections, 2017: an analysis for the Global Burden of Disease Study 2017. Lancet Respir Med. 2019 Jan 1;7(1):69–89. 

11. Effati F, Karimi H, Yavari A. Investigating effects of land use and land cover patterns on land surface temperature using landscape metrics in the city of Tehran, Iran. Arab J Geosci. 2021 Jun 24;14(13):1240. 

12. Karimi B, Shokrinezhad B. Air pollution and the number of daily deaths due to respiratory causes in Tehran. Atmos Environ. 2021 Feb 1;246:118161.

---

## [Decision Letter · Decision Letter 1]

9 Nov 2022

PONE-D-22-08786R1Geospatial epidemiology of hospitalized patients with a positive influenza assay: A nationwide study in Iran, 2016-2018PLOS ONE

Dear Dr. Kiani,

Thank you for submitting your manuscript to PLOS ONE. After careful consideration, we feel that it has merit but does not fully meet PLOS ONE’s publication criteria as it currently stands. Therefore, we invite you to submit a revised version of the manuscript that addresses the points raised during the review process.

We look forward to receiving your revised manuscript.

Kind regards,

Hamid Sharifi

Academic Editor

PLOS ONE

Additional Editor Comments (if provided):

Dear Authors,

The main concern on this submission is the quality of the reported data. Please review the comment and send your response about this important comment.

Best Regards

Reviewers' comments:

Reviewer's Responses to Questions

**Comments to the Author**

1. If the authors have adequately addressed your comments raised in a previous round of review and you feel that this manuscript is now acceptable for publication, you may indicate that here to bypass the “Comments to the Author” section, enter your conflict of interest statement in the “Confidential to Editor” section, and submit your "Accept" recommendation.

Reviewer #1: All comments have been addressed

Reviewer #2: All comments have been addressed

Reviewer #3: (No Response)

2. Is the manuscript technically sound, and do the data support the conclusions?

Reviewer #1: Partly

Reviewer #2: Yes

Reviewer #3: (No Response)

3. Has the statistical analysis been performed appropriately and rigorously? 

Reviewer #1: Yes

Reviewer #2: Yes

Reviewer #3: (No Response)

4. Have the authors made all data underlying the findings in their manuscript fully available?

Reviewer #1: Yes

Reviewer #2: Yes

Reviewer #3: (No Response)

5. Is the manuscript presented in an intelligible fashion and written in standard English?

Reviewer #1: Yes

Reviewer #2: Yes

Reviewer #3: (No Response)

6. Review Comments to the Author

Reviewer #1: My previous recommendation about this paper was rejection. This decision was different with editor and other reviewers. Now, I agree with editor decision

Reviewer #2: The manuscript has been greatly improved and the authors have addressed all of my comments. I don't have any further comments.

Reviewer #3: (No Response)

7. PLOS authors have the option to publish the peer review history of their article (what does this mean?). If published, this will include your full peer review and any attached files.

Reviewer #1: No

Reviewer #2: No

Reviewer #3: No

---

## [Author Response · Author response to Decision Letter 1]

23 Nov 2022

Dear Dr. Hamid Sharifi;

I appreciate you and the reviewers for examining our manuscript again. I would like to emphasize that we had mentioned in different places in our manuscript that our data is the representative of hospitalized PCR-confirmed cases of influenza, and we had not claimed that this is a total flu representative in the country. For instance, if one checks our manuscript title, our study subjects are those hospitalized patients with PCR-confirmed tests of Influenza. As a result, we have not claimed that our study is the whole picture of total Influenza or Influenza-Like Illness in the country. That would be another study with a different dataset which is the reviewer’s concern. We have responded to the reviewer’s comment as follows. Also, some revisions have been performed according to the comment.

Sincerely, 

On behalf of all authors,

Dr. Behzad Kiani

École de Santé Publique de L’Université de Montréal (ESPUM), Québec, Montréal, Canada. Email: kiani.behzad@gmail.com

Reviewer's comments:

Reviewer #3: 

Reviewers’ comment: As already mentioned, my comments are methodological in nature; and the fundamental objection to the nature of the data used in this study. In other words, based on the usual implementation method in hospitals, the data of this article is a “non-representative sample”. This problem has not been noticed by the authors, so that neither the main findings, nor the short title, nor the abstract of the article mention this important issue. In addition, the authors clearly concluded that influenza hotspot clusters have been identified. Considering the fact that this study’s data is a non-representative sample, this claim is incorrect. Considering all the above-mentioned, I suggest to the editor to reject the manuscript. But if the editor's opinion is not like this, at least the authors should be asked to add the number of PCR tests performed in each province during the study period. It can be used as bases for comparison and to understand why the claim of identifying influenza hotspot clusters is incorrect.

Response: Thanks for your comment and your time in examining our manuscript again. We respect your concern, but we have not claimed that our study is representative of total influenza in the country. As the study’s title shows, our study is spatial epidemiology of hospitalized patients with PCR-confirmed tests in Iran. So, when a reader reads our title, it is clear that our study is different from total influenza in terms of two limitations: first, all patients were hospitalized, the second, all of them were confirmed by PCR test. Also, we had mentioned in the Abstract > Introduction and Abstract > Methods that our study scope is related to hospitalized and PCR-confirmed cases. We think when somebody reads the title and abstract, it is absolutely clear what our study subjects are. However, according to your comment and to clarify our study scope better for the readers, we have revised our manuscript as follows:

1- The short title has been revised as follows:

Geospatial epidemiology of lab-confirmed hospitalized influenza patients.

2- Also, the Abstract > Conclusion has been revised as follows:

We characterized the spatial and epidemiological heterogeneities of severe hospitalized influenza cases confirmed by PCR in Iran. Detecting influenza hotspot clusters could inform prioritization and geographic specificity of influenza prevention, testing, and mitigation resource management including vaccination planning in Iran. 

3- We have revised the limitation part as follows:

We acknowledge that the results of this study are likely “non-representative” of the entire population.

4- According to your request, we have uploaded the PCR data for each province via S1 Appendix.

We think these clarifications are enough because it is not concise to write hospitalized and PCR-confirmed influenza cases everywhere in the manuscript when we have mentioned it many times before.

---

## [Editor Report · Decision Letter 2]

25 Nov 2022

Geospatial epidemiology of hospitalized patients with a positive influenza assay: A nationwide study in Iran, 2016-2018

PONE-D-22-08786R2

Dear Dr. Kiani

We’re pleased to inform you that your manuscript has been judged scientifically suitable for publication and will be formally accepted for publication once it meets all outstanding technical requirements.

Kind regards,

Hamid Sharifi

Academic Editor

PLOS ONE
---

## [Editor Report · Acceptance letter]

1 Dec 2022

PONE-D-22-08786R2 

Geospatial epidemiology of hospitalized patients with a positive influenza assay: A nationwide study in Iran, 2016-2018 

Dear Dr. Kiani:

I'm pleased to inform you that your manuscript has been deemed suitable for publication in PLOS ONE. Congratulations! Your manuscript is now with our production department. 

Kind regards, 

on behalf of

Dr. Hamid Sharifi 

Academic Editor

PLOS ONE